# Determination of Five Sit-to-Stand Test Performance at Discharge of Stroke Patients

**DOI:** 10.3390/diagnostics14050521

**Published:** 2024-02-29

**Authors:** Maria Piedad Sánchez-Martínez, María José Crisostomo, Rodrigo Martín-San Agustín, Joaquina Montilla-Herrador, María Pilar Escolar-Reina, Elisa Valera-Novella, Francesc Medina-Mirapeix

**Affiliations:** 1Department of Physiotherapy, University of Murcia, 30100 Murcia, Spain; mariapiedad.sanchez1@um.es (M.P.S.-M.); montilla@um.es (J.M.-H.); pescolar@um.es (M.P.E.-R.); elisa.valeranovella@gmail.com (E.V.-N.); mirapeix@um.es (F.M.-M.); 2Department of Rehabilitation, Jerez Hospital, 11407 Jerez de la Frontera, Spain; mjcriso@gmail.com; 3Clinimetry and Technological Development in Therapeutic Exercise Research Group (CLIDET), Department of Physiotherapy, University of Valencia, 46010 Valencia, Spain

**Keywords:** stroke, sit-to-stand test, prediction, performance, outpatient rehabilitation program, discharge

## Abstract

The early identification of performance in the five-repetition sit-to-stand test (5-STS) at discharge in stroke patients could be of interest because it can determine independence for community-based activities. This study aimed to determine whether the initial measurement of the 5-STS test can be a determinant of the performance level prediction and amount of change in the 5-STS test at discharge in stroke patients. A prospective cohort study was conducted with a sample of 56 patients aged ≤60 d post-stroke. The 5-STS test results, as well as changes in patient condition, were measured at admission (T0) to an outpatient rehabilitation program, after the first month (T1), and at discharge (T2). The mean age was 62.7 (SD = 13.0), 58.9% of the subjects were male, and 75% had suffered an ischemic stroke. A multivariate linear regression model using the 5-STS test at T0 explained 57.7% of the variance in the performance at discharge. Using the 5-STS at T1 increased the variance to 75.5% (*p* < 0.001). Only the time from stroke onset at T0 significantly contributed to the two models. The measurement of the 5-STS at T0 and the amount of change in its performance at T2 explained 60.2% (*p* < 0.001) of the variance, while reassessment at T1 explained only 19.3% (*p* < 0.001). The level of patient performance on the 5-STS test at discharge, as well as any potential change, can be predicted by the admission measure of 5-STS in stroke patients.

## 1. Introduction

Unfortunately, about a third of stroke patients are permanently disabled and require assistance with daily activities [1]. Sit-to-stand (STS) from a chair is one of the most affected functional tasks after a stroke [2], and key to independence for community-based activities [3]. The ability to perform this movement safely, and independently can be influenced by a range of common post-stroke impairments, including muscle weakness, balance problems, or poor functional status, among others [4,5,6]. Although some patients can perform this movement, many of them do not perform it with a fast-enough speed, making it a very physically demanding movement [7]. Consequently, recovery of STS movement is one of the primary goals after a stroke [7].

Current evidence shows that treatment and training interventions specifically aimed at improving STS may be effective in improving the time to perform the STS a specified number of times, in people who can perform this movement after a stroke [8]. Despite the effectiveness of these specific interventions, there is notable variability in the number of times or in the time to perform STS between patients. For example, the study of Janssen et al. showed a coefficient of variation (CV) of 91.8% for the number of STS repetitions during the first 3 months of functional recovery after a stroke [9]. Although patients receive therapies aimed at improving STS, they show substantial variability in treatment response [8,9,10].

The knowledge of the predictors of treatment response could be used to better match patients with an effective treatment [10]. Given the need to optimize resources, it would be useful to investigate whether any factor available at the start of rehabilitation may have a systematic relationship with either the level of performance achieved after 4 weeks and the end of the rehabilitation or the amount of change during these periods. Previous studies have identified that many physical improvements depend on the baseline function [11,12]. Therefore, we hypothesize that STS at baseline after a stroke could be used as a predictor of the performance level and the amount of change of it at discharge.

To provide an answer to this hypothesis, the five sit-to-stand (5-STS) test could be used, since this test measures the time required to stand five times from a sitting position as quickly as possible [13]. The 5-STS test is reliable in individuals following a stroke [14], has a potential link with falls [15], and has better clinimetric properties than assessments with set time limits (e.g., 30 s chair stand) [16]. Given its applicability in clinical practice, this measure could be used to improve our understanding of the recovery of STS-related functioning after an acute stroke and treatment response.

The aims of the present study were: (1) to determine if the initial measurement of the 5-STS can be a determinant of the prediction of the achieved level of performance in the 5-STS test at discharge, and the amount of change in the 5-STS test from the initial measurement to discharge for stroke patients undergoing an outpatient rehabilitation program; (2) to determine if an early reassessment at 30 days can improve the predictive ability of the admission score; and (3) to investigate whether the prediction of the achieved level and amount of change in the 5-STS at discharge was enhanced by the addition of other variables.

## 2. Materials and Methods

### 2.1. Study Design

A cohort study was conducted which included patients with a first episode of stroke who were admitted to the Rehabilitation Service of the Jerez Hospital (Spain) between October 2016 and October 2018, and these patients were invited to participate in an outpatient rehabilitation program (ORP) after receiving acute stroke care at that or a nearby hospital. Assessments were made by hospital nursing management staff within two days of admission (T0), repeated at 4 weeks (T1), and at discharge. The study protocol was approved by the Ethical Committee of the Jerez Hospital (approval number: EST-42/16; approval date: 26 July 2016).

### 2.2. Participants

Participants were prospectively recruited and screened. The inclusion criteria included patients who were at least 30 years of age, had relatively good functional status in order to be able to walk at least 4 m, were able to complete the 5-STS test, and were screened within two months post-stroke. The patients were excluded when they displayed cognitive or language impairments (in the subscales of the Cognistat) [17] and any physical impairments that would prevent them from performing the 4 m walk test or the 5-STS (for example, knee problems). The cognitive and language criteria were based on the need to ensure patients could provide reasonable answers. Additionally, due to the second study aim, a withdrawal criterion was applied for those patients who were discharged before the first follow-up visit (T1). All study patients provided written informed consent.

### 2.3. Intervention

The ORP was individualized and personalized, adapting to the patient’s evolution, by nursing staff. This ORP included interventions aimed at improving the ability to sit-to-stand independently, such as repetitive practice of sit-to-stand and of the components required for movement from sitting to standing: muscle strength training and provision of feedback.

### 2.4. Outcome Measures and Predictors

The outcome measures were the time in the 5-STS test at discharge, and the changes in this test from T0 and T1 to discharge.

The primary predictors were the 5-STS test results at T0 and at T1. This test has shown validity, reliability, and responsiveness [14,17]. The 5-STS test was measured as the time taken to complete five repetitions of the sit-to-stand task. All sit-to-stand tasks were performed using a chair without an armrest, with a height of 43 cm and a depth of 47.5 cm. Timing began when the patient’s back left the backrest and stopped once the back touched the backrest for the fifth time [13]. In addition, several secondary predictors, accessible at the time of admission and associated with functional changes during rehabilitation, were included: age (years), sex, side of brain damage (right/left), diagnosis (ischemic or hemorrhagic stroke), the time interval between stroke onset and first assessment (T0), the Functional Ambulatory Category (FAC), and gait speed [12,18,19]. The FAC was used for the walking ability, which is a clinical and visual gait assessment scale that distinguishes between 6 levels of walking ability based on the amount of physical support required, from level 0 (the patient cannot walk at all or needs the help of 2 therapists) to level 5 (the patient can walk everywhere independently) [11]. Gait speed was assessed with a 4 m gait speed test and reported in m/s. Subjects were asked to complete the 4 m walk at their “most comfortable speed,” and a stopwatch recorded the time. Timing began after an acceleration distance of 1 m and ceased as they crossed a finish line. Subjects performed two trials, and the faster time was recorded [17].

### 2.5. Statistical Analysis

Participants’ characteristics and functioning at admission were summarized for the whole sample, and functioning was again described at T1 and discharge. Categorical variables were expressed as frequencies and proportions (%) and continuous variables as mean (±standard deviation) or median (interquartile range [IQR]) as appropriate. In addition, a one-way repeated measures ANOVA was also conducted to compare performance on the 5-STS test across the three time points. Moreover, we performed multiple post hoc comparison tests to determine which time points were significantly different from others, using the Bonferroni multiple-significance-test correction.

Simple linear regressions were carried out to assess the 5-STS test results at admission and T1 and each one of the secondary predictors, using its performance and its change at discharge as a dependent variable. The variables that showed a statistically significant association (*p* < 0.10) with the measures of the 5-STS test at discharge were entered into a multivariate linear model. All models were produced using the enter method.

Sample size calculation was based on the general rule that 15 subjects per predictor are needed for a reliable equation [20]. We recruited a minimum of 56 participants, assuming a maximum of three predictors. All analyses were conducted using SPSS v.24 (IBM SPSS, Chicago, IL, USA).

## 3. Results

### 3.1. Patient Characteristics at Admission, T1, and Discharge

In total, 56 stroke patients were initially included. All of them, except one (due to death), participated during the study. The mean age was 62.7 (SD = 13.0), 58.9% of the subjects were male, and 75% had suffered an ischemic stroke. The mean time since stroke onset was 48.4 days (SD = 30.3). The median of time to discharge was 57 days (IQR = 50). Table 1 summarizes relevant patients’ characteristics and the means (SD) of performance on the 5-STS test at admission, T1, and discharge.

A mean performance on the 5-STS test was 16.1 s (SD = 6.72) at T0, 13.2 s (SD = 4.71) at T1, and 11.6 s (SD = 4.2) at discharge. The one-way repeated measures showed that the performance on the 5-STS test differed significantly across the three time points (F = 47.7; *p* < 0.001), and the Bonferroni *t*-test, which was also different among these three time points, also reached significance (*p* < 0.001). Figure 1 shows that changes from admission to discharge in the 5-STS test had a median of 3.3 s (IQR = 4.7), and these statistics had lesser values for changes after the first month (Median = 0.9, IQR = 2.9). All the outliers were patients with high times on the 5-STS test at admission (≥25 s).

### 3.2. Predictive Ability of the 5-STS Test Performance at Discharge

A simple linear regression to predict 5-STS test performance at discharge using admission 5-STS test measures produced a significant model in which the variance explained was 47.6%, F(1.53) = 55.24 (*p* < 0.001). In this model, the 5-STS test at admission significantly predicted the performance at discharge (β = 0.437, *p* < 0.001). The final predictive model was 5-STS at discharge = 4.55 + (0.44 × 5-STS at baseline). For the 5-STS measures at the first month, simple regression also found a significant model, but it notably increased the variance explained to 72.6%, F(1.53) = 679.9 (*p* < 0.001). In this model, the 5-STS test was also statistically significant (β = 0.769, *p* < 0.001). Figure 2 shows the scatter plot and equation of this simple regression model for the 5-STS test measurements at the first month and at discharge.

Table 2 reports the results of the two multiple regression models predicting the 5-STS test performance at discharge on each one of the 5-STS test measures (at admission and first month) and the selected covariates (Appendix A). The two models explain the 57.7% and 75.5% of the variance, respectively, which represents a slight increase regarding their respective simple regression models (2.5–10%). While the time from stroke to admission to ORP contributed significantly to the two models, age did not. Nevertheless, the partial correlation for time from stroke to admission to ORP was found to have notably lower values than those from the 5-STS test measurements. For example, in Model 1, while the 5-STS test at admission (T0) explained 27.04% (0.5202) of the variance in the total R2 of its model, time from stroke to admission to ORP uniquely explained 9% (0.3002). The plot of residuals did not suggest that the assumptions of linearity, homoskedasticity, and normality were problematic.

### 3.3. Predictive Ability for the Change in the 5-STS Test Results at Discharge

Simple linear regression confirmed that the relationship between baseline 5-STS measures and the amount of change in performance at discharge was significant (β = 0.563; 95%CI = 0.43–0.68; *p* < 0.001). This prediction model explained 60.2%, F(1.53) = 81.54 (*p* < 0.001), of the variances for changes in the 5-STS test results between admission and discharge. Figure 3 shows the scatter plot between that relationship and the regression equation. As can be seen, patients who took longer to perform the 5-STS test at admission changed more than those with higher performance (performed the test faster). For example, patients who performed the initial 5-STS test between 30 and 35 s achieved a greater change ranging between 12–22 s (except for one patient) than those who performed the tests below 15 s, where the mean change was 5 s or they did not change.

For the 5-STS measurements at the first month, although the simple regression was significant (β = 0.231; 95%CI = 0.10–0.36; *p* = 0.001), the explained variance experienced a notable decrease to 19.3%, F(0.23) = 12.9 (*p* = 0.001). On the other hand, the amount of change from T0 to T1 and performance at discharge from the 5-STS showed no association (β = 0.174; 95% CI = −0.12–0.46; *p* = 0.242).

No multiple regression, including any secondary predictors, was carried out to test the predictive abilities of the 5-STS test change score because none of the results were initially statistically significant (Appendix A).

## 4. Discussion

The present study showed that the assessment of the 5-STS test at the start of an outpatient rehabilitation program allows for predicting the achieved level of performance and the amount of change of the 5-STS test at discharge of these programs. This study also showed that reassessment of the 5-STS test after 30 days improves the prediction of the performance level of the 5-STS test at discharge; however, the prediction of the amount of change worsens.

To our knowledge, this is the first study demonstrating that the performance level of the 5-STS test at discharge can be predicted from the admission assessment. Previous studies have only reported the cross-sectional ability of the 5-STS test for discriminating patients with differing ambulatory capacity, but not its relationship with the state at discharge [18]. Nevertheless, several studies have shown that the initial measurement after a stroke may be relevant when it comes to determining the status at discharge of these patients. Goldie et al. demonstrated in their study that admission measurements, in this case gait speed, can predict patient discharge status with a high percentage of explained variance [11]. Our study also shows that the 5-STS test measurement on admission explains a high percentage of the variance of the model, which only improves with the time from stroke to admission to ORP but with a low contribution. As we expected, the time since the stroke is a significant covariate in this model, given that time-related progress has been shown to be an important factor for post-stroke recovery. However, previous studies have shown that, when considering the time variable in isolation, it only explains 16% of the improvements in post-stroke body functions [21]. This contribution of time to the model is usually mainly attributed to the first few weeks since the recovery after stroke and shows a non-linear logarithmic pattern [22], where the greatest improvements are observed early after the onset of stroke, which gradually stabilizes later [23,24]. On the other hand, our finding that age did not add accuracy to the model was not surprising. Jorgensen et al. previously indicated in their study that post-stroke motor recovery was strongly influenced by the degree of disability of the lower limbs at admission beyond a certain age, which could explain why age was not a determinant for this model [23].

Our study also showed that an early reassessment of the 5-STS test results allows for obtaining a more accurate prediction of its performance level at discharge, with an increase in variance of 25% concerning the initial measurement. Previous studies have already identified a positive association between the evaluation at 1 month of the sit-to-stand position and the state at discharge, and our study confirms this association [25]. This increase in the variance with respect to the initial measurement could be due to the fact that the 5-STS test requires a great effort on the part of the patient, as well as some balance and strength of the lower limbs, which are more affected on admission than during the first month after admission [6,25]. As the literature indicates, the greatest recovery of motor functions occurs during the first 4 weeks post-stroke, which could explain why the one-month reassessment has the potential to be a more realistic reflection of the performance on the 5-STS test results at discharge than the measure upon admission [22].

A relevant finding of this study was that the admission measure of the 5-STS test has been shown to predict the amount of change in the performance at discharge with an accuracy of 60% after rehabilitation. This means that patients with lower performance in the 5-STS test on admission have a wider range of change than patients with better performance, as Vanclay et al. suggest in their study [26]. Therefore, expectations about the change in performance of the 5-STS test at discharge of stroke patients can be based on the state at the start of rehabilitation. This finding has important clinical consequences since it will allow us to determine the effectiveness of rehabilitation. Although Vanclay et al. identified the amount of improvement averaged over the duration of the rehabilitation, it was related to the efficiency of the rehabilitation [26]. In contrast, the early reassessment of the 5-STS test results did not accurately predict the condition at discharge. These findings are not surprising since, as mentioned above, motor changes occur during the first month; therefore, the patients will present a better functional state and their capacity for change will be much less than at the start [23]. Similarly to Goldie et al. in their study of gait speed, none of the covariates analyzed were related to the amount of change in the performance of the 5-STS test during the rehabilitation period [11].

### 4.1. Implications for Practice and Research

Our study has evidenced that an assessment of the 5-STS test at admission and an early reassessment at 30 days allows an accurate prediction of the level of performance of the 5-STS test at outpatient discharge from the facilities. That finding is important for clinicians to support tailored interventions for these patients with a poor prognostic and to advise them and/or their families about the prognostic [22]. Moreover, early reassessment could prove to be interesting because, at the moment of admission, the patient may be disoriented by being in a hospital instead of at home; thus, the patient may be overwhelmed by the present situation and, therefore, may not respond well to requests [25]. All these situations may not represent the initial actual state of the patient and could make the rehabilitation treatment non-focused according to the possibilities of the patient, thus slowing down the recovery process. In addition, the initial 5-STS test measurement can also predict the amount of change at discharge. This information is essential, not only for the rational planning of medical care, but also for the reliable prognosis of the time course of recovery and the duration of rehabilitation in individual patients characterized by the initial stroke severity. The predictive ability of the 5-STS test results should open up the opportunity for new research with which to evaluate a candidate for monitoring disease progression and predicting differential treatment responses to interventions [27].

### 4.2. Limitations

The present study is subject to some limitations. First, as our study was developed at a single hospital and included a small number of patients, generalizations should be made cautiously. To improve the external validity of these results, this study should be replicated in a more diverse population across multiple centers. Second, we selected patients who were able to walk 4 m and perform the 5-STS test upon admission to the rehabilitation program. This aspect limits its applicability to patients with relatively good functional status; hence, future research should be aimed at establishing whether the results found in this study would be applicable to a population with greater functional impairment. Third, other measures that may impact improvement during rehabilitation and also concern the 5-STS test were not included as covariates. For example, lower extremity impairment (somatosensorial, spasticity, and muscle strength) was not specifically measured, yet may impact 5-STS test *n* [28].

## 5. Conclusions

In conclusion, the initial assessment of 5-STS is a determinant of the prediction of the achieved level of performance and the amount of change from 5-STS at discharge for stroke patients undergoing outpatient rehabilitation programs. Early reassessment at 30 days improves the predictive ability of the admission score only for the level of performance of the 5-STS test at discharge. Further studies are required to determine the prognostic value of this test for monitoring disease progression and to improve the effectiveness of rehabilitation treatment.

## Figures and Tables

**Figure 1 diagnostics-14-00521-f001:**
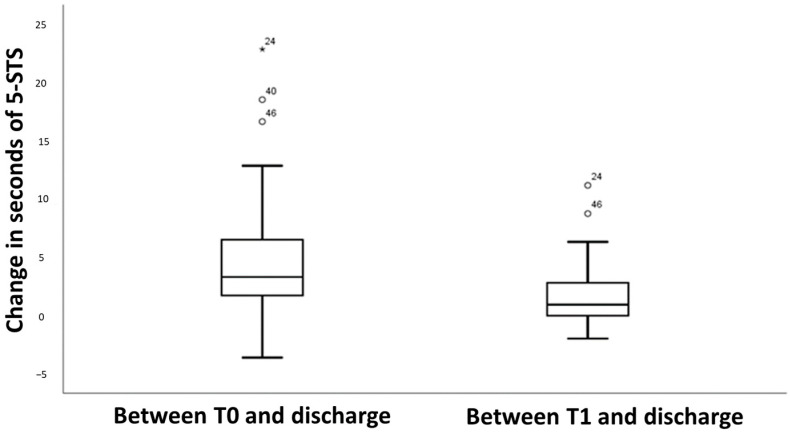
Box plots showing the changes of performance in the 5-STS test between admission (T0) and T1 following discharge.

**Figure 2 diagnostics-14-00521-f002:**
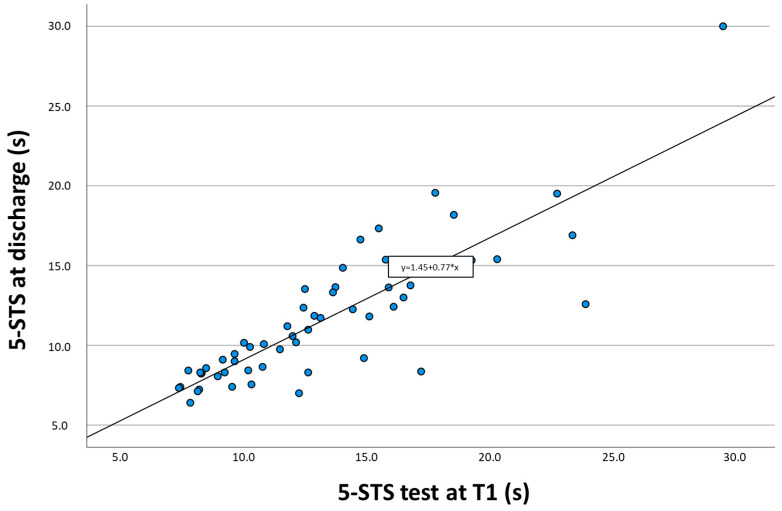
The scatter plot shows the relationship between performance of 5-STS at first month and level of performance in 5-STS at discharge.

**Figure 3 diagnostics-14-00521-f003:**
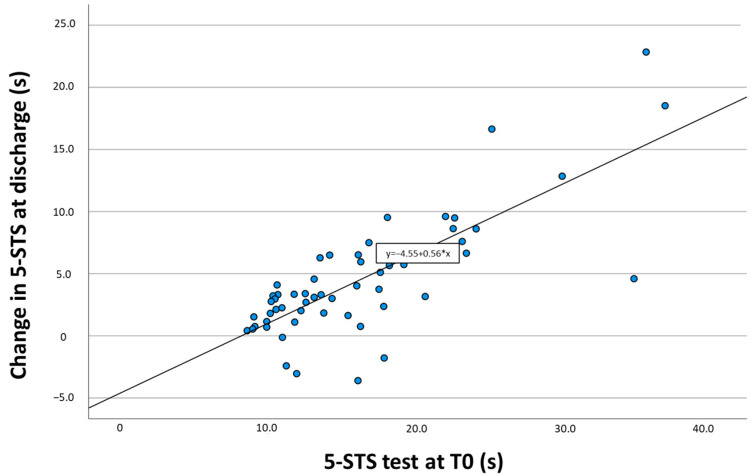
The scatter plot shows the relationship between performance of the 5-STS test at admission (T0) and change in 5-STS test results between T0 and discharge.

**Table 1 diagnostics-14-00521-t001:** Baseline demographics and overall stroke patient characteristics of study population.

Variable	All (*n* = 56)
Demographics	
Age. years; mean (SD)	62.7 (13.0)
Gender (male)	33 (58.9%)
Stroke characteristics	
Side affected (right)	27 (48.2%)
Type of stroke (ischemic)	42 (75.0%)
Time from stroke to admission ORP. days; mean (SD)	48.4 (30.3)
Time in ORP. days; mean (SD)	70.7 (40.4)
FAC (score 0–5), median (IQR)	4.0 (1.0)
Gait speed. m/s; mean (SD)	0.62 (0.32)
5-STS at T0. s; mean (SD)	16.10 (6.72)
5-STS at T1. s; mean (SD)	13.2 (4.7)
5-STS at discharge. s; mean (SD)	11.6 (4.25)

Data are presented as mean ± standard deviation, median (interquartile range), or number (percentage %); SD indicates standard deviation; ORP, outpatient rehabilitation program; FAC, Functional Ambulation Category; IQR, interquartile range; 5-STS, five-repetition sit-to-stand.

**Table 2 diagnostics-14-00521-t002:** Multivariate prediction of performance level in the 5-STS test at discharge.

Variable	Multivariate Models		
	B (95%CI)	*p*-Value	R2
Model 1			0.577
5-STS at T0	0.36 (0.23–0.48)	<0.001	
Age, y	0.03 (−0.02–0.09)	0.249	
Time from stroke to admission ORP, days	0.04 (0.01–0.07)	0.002	
Model 2			0.755
5-STS at T1	0.69 (0.54–0.83)	<0.001	
Age, y	0.01 (−0.03–0.06)	0.512	
Time from stroke to admission ORP, days	0.02 (0.00–0.04)	0.019	

5-STS indicates five-repetition sit-to-stand; ORP, outpatient rehabilitation program, CI, confident interval; R2, variance.

## Data Availability

Data is contained within the article and Appendix A.

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
