# Peer review of "Determination of Five Sit-to-Stand Test Performance at Discharge of Stroke Patients"

_diagnostics, 2024, doi:10.3390/diagnostics14050521_

Round 1

Reviewer 1 Report

Comments and Suggestions for Authors

1. Sample Selection Criteria: The study selects patients who can walk 4 meters and perform the 5-STS test, indicating they have relatively good functional status. This selection could bias the results towards better-performing patients and does not represent patients with poorer functional status. The criteria for patient selection should be broadened or clearly justified.

2. Inclusion of Other Rehabilitation Measures: The study does not include measures such as lower extremity impairment (e.g., somatosensory, spasticity, muscle strength) which may impact 5-STS test results. Including these measures could provide a more comprehensive understanding of factors influencing sit-to-stand performance.

* The study is conducted at a single hospital with a relatively small sample size. This limitation might affect the generalizability of the findings. Future studies should involve a more diverse patient population across multiple centers to enhance the external validity of the results.

* The study focuses primarily on the 5-STS test without considering other potential factors that might influence stroke recovery, such as cognitive function, nutrition, or psychological state. Including these factors could provide a more holistic understanding of stroke recovery.

The study assesses patients at discharge, which provides short-term outcomes. Including long-term follow-up assessments could help in understanding the sustained effects of rehabilitation and the predictive power of the 5-STS test over a longer period.

Author Response

Response to Reviewer 1 Comments

We want to thank reviewer 1 for providing us with suggestions to improve our manuscript.

Point 1. Reviewer’s comment: Sample Selection Criteria: The study selects patients who can walk 4 meters and perform the 5-STS test, indicating they have relatively good functional status. This selection could bias the results towards better-performing patients and does not represent patients with poorer functional status. The criteria for patient selection should be broadened or clearly justified.

Response 1: Thank you very much for your appreciation. This aspect is one of the limitations of our study and therefore one must be careful when extrapolating our results to patients who are unable to walk or have a poor functional status before admission to the rehabilitation program. For your better understanding, this aspect has been clarified in the inclusion criteria.

Point 2. Reviewer’s comment: Inclusion of Other Rehabilitation Measures: The study does not include measures such as lower extremity impairment (e.g., somatosensory, spasticity, muscle strength) which may impact 5-STS test results. Including these measures could provide a more comprehensive understanding of factors influencing sit-to-stand performance.

Response 2: We greatly appreciate your indication. We agree with his comment and it would be a future line of research. We show this aspect in the manuscript as a limitation of our study, since we could not measure certain aspects as relevant as muscle strength.

Reviewer 2 Report

Comments and Suggestions for Authors

This paper reports on the results of a single centre prospective cohort study of the predictive value of the baseline (T0) and month (T1) 1 five-repetition sit-to-stand (5-STS) timings with the timings at discharge (T2) and timings changes after an out-patient rehabilitation program. The study showed correlations of 5-STS at T0 and even better T1.

There are some issues that authors may wish to address:

1.       Language issues appear through-out the paper, affecting readability eg grammar, vocabulary, punctation, sentence structure

2.       ‘5STS’ and ‘5-STS’ seem to appear in the paper; there is also inconsistent use of T0, T1 and T2 eg ‘discharge’ used instead of ‘T2’ – please standardise

3.       Abstract – to add some demographic data eg age, gender, stroke types, time from stroke, time in the program, etc. Explain T0, T1. T2 at first use

4.       Lines 22-23 seem to contradict lines 20-21 – please make it clear what is being highlighted here

5.       Line 80 – were those with significant knee or spine problems excluded as this may affect their 5-STS?

6.       Line 95 – what about FAC and gait speed? These terms suddenly appeared in Table 1. How were they measured?

7.       Table 1 - what about number of days of ORP? I do not see T1 and T2 5-STS timings, No ‘SD’ in the table

8.       Fig 3 - why not also show a plot of T0 vs T2?

9.       Fig 3 – is it that those which higher 5-STS at T0 had further increase in 5-STS at discharge (ie deterioration)?; and those with lower had reduction? I suppose the ‘-‘ sign on the y-axis to signify reduction in 5-STS ie improvement?

10.   Would a better improvement in 5-STS from T0 to T1 will be better correlated with a better 5-STS at T2?

Comments on the Quality of English Language

Needs attention - Language issues appear through-out the paper, affecting readability eg grammar, vocabulary, punctation, sentence structure

Author Response

Response to Reviewer 2 Comments

We want to thank reviewer 2 for providing us with suggestions to improve our manuscript.

Point 1. Reviewer’s comment: Language issues appear through-out the paper, affecting readability eg grammar, vocabulary, punctation, sentence structure.

Response 1: We agree, thank you again for your indications. The manuscript has been reviewed again by an expert to correct typographical and grammatical errors.

Point 2. Reviewer’s comment: ‘5STS’ and ‘5-STS’ seem to appear in the paper; there is also inconsistent use of T0, T1 and T2 eg ‘discharge’ used instead of ‘T2’ – please standardise.

Response 2: Thank you very much for your comment, this aspect has been modified in the manuscript.

Point 3. Reviewer’s comment: Abstract – to add some demographic data eg age, gender, stroke types, time from stroke, time in the program, etc. Explain T0, T1. T2 at first use.

Response 3: We agree, thank you again for your indications. This aspect has been added in the manuscript, see lines 18-19.

Point 4. Reviewer’s comment: Lines 22-23 seem to contradict lines 20-21 – please make it clear what is being highlighted here.

Response 4: Thank you very much for your comment, this aspect has been clarified in the manuscript. In lines 20-21 it refers to the model in which the 5-STS measurement is used in T0 and T1. While the model referred to in lines 22-23 uses the 5-STS measurement at T0 and amount of change in its performance at discharge.

Point 5. Reviewer’s comment: Line 80 – were those with significant knee or spine problems excluded as this may affect their 5-STS?

Response 5: Thank you very much for your comment. Only patients who will be able to take the 4-meter walk test and the 5-STS are included. This aspect has been clarified in the exclusion criteria.

Point 6. Reviewer’s comment: Line 95 – what about FAC and gait speed? These terms suddenly appeared in Table 1. How were they measured?

Response 6: Thank you very much to your comments. We have added this aspect in the manuscript. Please see the lines 110-118.

Point 7. Reviewer’s comment: Table 1 - what about number of days of ORP? I do not see T1 and T2 5-STS timings, No ‘SD’ in the table.

Response 7:  We have included these aspects in table 1.

Point 8. Reviewer’s comment: Fig 3 - why not also show a plot of T0 vs T2?

Response 8: Thank you very much for your appreciation. Figure 3 shows the relationship between the 5-STS measurement at admission and the amount of change in its performance at discharge, therefore the relationship between the initial measurement of T0 and performance at discharge (T2) is shown.

Point 9. Reviewer’s comment: Fig 3 – is it that those which higher 5-STS at T0 had further increase in 5-STS at discharge (ie deterioration)?; and those with lower had reduction? I suppose the ‘-‘ sign on the y-axis to signify reduction in 5-STS ie improvement?

Response 9: Thank you very much for your comment. This aspect has been clarified in the manuscript in lines 201-202. " As can be seen, patients who took longer to perform the 5-STS test at admission changed more than those with higher performance (performed the test faster)."

​Point 10. Reviewer’s comment: Would a better improvement in 5-STS from T0 to T1 will be better correlated with a better 5-STS at T2?

Response 10: Thank you very much for your comment. This aspect was previously analyzed and there was no correlation between the change from T0 to T1 and the situation at discharge.

Round 2

Reviewer 1 Report

Comments and Suggestions for Authors

One area that could benefit from further review and revision is the study's limitation section. While the authors acknowledge the study's single-hospital context and the relatively good functional status of the included patients, expanding this section to address the potential for selection bias and its implications on the generalizability of the findings could enhance the paper. Specifically, discussing how the inclusion criteria may limit the applicability of the results to a broader stroke patient population and exploring strategies for future research to include more diverse patient groups would provide a more nuanced understanding of the study's applicability and limitations.

Author Response

Response 1: Thank you very much for your comment. For better understanding we have added your suggestion into the manuscript in the limitations section.

Reviewer 2 Report

Comments and Suggestions for Authors

This paper is a revised submission on the results of a single centre prospective cohort study of the predictive value of the baseline (T0) and month (T1) 1 five-repetition sit-to-stand (5-STS) timings with the timings at discharge (T2) and timings changes after an out-patient rehabilitation program.

The paper is much improved, but some issues remain inadequately addressed:

1.       Language issues – I have no time to advise on the details - I will appreciate the help of the publisher to advise the authors - see my comments 3 and 4 below as examples….

2.       Abstract – the newly added demographic data (lines 18-19) should appear after the sentence on “The 5-STS…’ lines 19-20. The terms T0 T1 T2 are not adequately utilised – I suggest that these be introduced in line 20, and be used for the rest of the abstract in place of terms such as ‘at discharge’ etc, except in the conclusion. The time to enter the program after the stroke and time in the program should be mentioned in the results portion of the abstract

3.       Line 113 – ‘then’ should be ‘they’?

4.       Lie 115 – ‘you’ should be ‘the participant?

5.       Table 1 – again about ‘SD’ in the table. I suspect the numbers in the brackets for the timings and speed are the SDs. If I am correct, please amend to eg. Time in ORP, days (SD)

6.       Point 10. Reviewer’s comment: Would a better improvement in 5-STS from T0 to T1 will be better correlated with a better 5-STS at T2? 

7.       Response 10: Thank you very much for your comment. This aspect was previously analyzed and there was no correlation between the change from T0 to T1 and the situation at discharge.

8.       My response – where was the previous analysis done? Which lines please?

Comments on the Quality of English Language

Still needs attention

Author Response

Response to Reviewer 2 Comments

We want to thank reviewer 2 for providing us with suggestions to improve our manuscript.

Point 1. Reviewer’s comment:    Language issues – I have no time to advise on the details - I will appreciate the help of the publisher to advise the authors - see my comments 3 and 4 below as examples….

Response 1: Thank you so much. The entire manuscript has been revised again and made multiple corrections and language changes.

Point 2. Reviewer’s comment: Abstract – the newly added demographic data (lines 18-19) should appear after the sentence on “The 5-STS…’ lines 19-20. The terms T0 T1 T2 are not adequately utilised – I suggest that these be introduced in line 20, and be used for the rest of the abstract in place of terms such as ‘at discharge’ etc, except in the conclusion. The time to enter the program after the stroke and time in the program should be mentioned in the results portion of the abstract

Response 2: Thank you very much for your suggestion. We have made the suggested changes.

Point 3. Reviewer’s comment: Line 113 – ‘then’ should be ‘they’?

Point 4. Reviewer’s comment:   Lie 115 – ‘you’ should be ‘the participant?

Response 3 and 4: Thank you so much. The entire manuscript has been revised again and made multiple corrections and language changes.

Point 5. Reviewer’s comment: Table 1 – again about ‘SD’ in the table. I suspect the numbers in the brackets for the timings and speed are the SDs. If I am correct, please amend to eg. Time in ORP, days (SD)

Response 5: Thank you very much for your suggestion. We have made the changes to the table following your comment.

Point 6. Reviewer’s comment: Point 10. Reviewer’s comment: Would a better improvement in 5-STS from T0 to T1 will be better correlated with a better 5-STS at T2?.

Response 10: Thank you very much for your comment. This aspect was previously analyzed and there was no correlation between the change from T0 to T1 and the situation at discharge.

 My response – where was the previous analysis done? Which lines please?

Response 6:  Thank you very much for your comment. This aspect has been incorporated into the manuscript, please see lines 213-215.